# *Candida auris*: A Continuing Threat

**DOI:** 10.3390/microorganisms13030652

**Published:** 2025-03-13

**Authors:** Ashish Bhargava, Katherine Klamer, Mamta Sharma, Daniel Ortiz, Louis Saravolatz

**Affiliations:** 1Thomas Mackey Center of Infectious Diseases, Henry Ford Health—St. John Hospital, Detroit, MI 48236, USA; 2School of Medicine, Wayne State University, Detroit, MI 48202, USA; 3LabCorp—Health Systems Operating Division, Troy, MI 48083, USA

**Keywords:** *Candida auris*, antifungal resistance, pathogenic fungi, nosocomial transmission

## Abstract

*Candida auris* is a World Health Organization critical-priority fungal pathogen that has variable resistance to antifungal treatments. Multiple clades have been identified through genomic analysis and have appeared in different geographic locations simultaneously. Due to a combination of factors including antifungal resistance, ability to colonize and persist in the environment, and thermotolerance, it can thrive. Infected patients are associated with a high mortality rate, especially those with multiple health risk factors like those associated with other *Candida* species. This review highlights the current situation of this pathogen to help provide guidance for future work.

## 1. Introduction

*Candida auris* (*C. auris*) is a multidrug-resistant fungal pathogen that poses a significant global health threat due to its ability to cause outbreaks in healthcare settings, resistance to disinfectants, and its persistence on human skin and in the environment [1]. The Centers for Disease Control and Prevention (CDC) recognized it as the first fungal pathogen classified as an urgent threat in their 2019 report [2]. Additionally, the World Health Organization (WHO) has designated it as a “critical” human fungal pathogen [3]. The first report of *C. auris* emerged in Japan in 2009, although it was retrospectively isolated as early as 1996 [4,5]. By 2018, cases were reported on all six inhabited continents, confirming its widespread presence [6]. Global outbreaks of this yeast are increasing, posing a significant threat due to its broad resistance to antifungal agents and its transmission in long-term-care facilities and hospitals [7].

In recent years, the rise in infections and colonization by non-albicans *Candida* species is believed to be linked to the excessive use of prophylactic antifungal agents, especially fluconazole [7]. Additionally, various factors contribute to the rising frequency of *C. auris* infections. These include climate change favoring thermotolerant yeasts, the fungus’s ability to spread among different species, potential gaps in infection control, the global shortage of healthcare workers and equipment, and the growing number of individuals with severe COVID-19 who are treated with antimicrobials [5,8].

Research on improving its identification and restriction in its spread by effective infection control measures by this opportunistic fungus must be prioritized urgently. This article provides a review of the epidemiology, at-risk populations, pathogenesis, clinical features, diagnostics, treatment, and infection control measures for *C. auris*.

## 2. Mycology

*C. auris* is a member of the ascomycetous (Hemi ascomycetes) *Clavispora* clade within the *Metschnikowiaceae* family, which is part of the order *Saccharomycetales*. It is a species of budding yeast that belongs to the genus *Candida*. Its name, “*auris*”, comes from the Latin word for “ear”. This fungus was first identified in the ear canal of a 70-year-old Japanese patient in 2009 [1]. The analysis of yeast genomic DNA identified a distinct species sharing close phylogenetic relationships with *C. ruelliae*, *C. haemulonii*, *C. duobushaemulonii*, and *C. pseudohaemulonii* [1,9]. *C. auris* is distinctive among *Candida* species because it rarely forms pseudohyphae or hyphae. It thrives at temperatures exceeding 40–42 degrees Celsius and in high-salinity environments [10]. *C. auris’s* ecological niche is not fully understood, but it appears to have originated as an environmental fungus in wetlands, thriving in various temperatures and salinity levels [11]. Changes related to global warming in specialized ecosystems, such as wetlands, may have significantly contributed to the emergence of *C. auris* as a human pathogen [12].

## 3. Epidemiology

*C. auris* isolates have revealed that this fungus emerged independently on several continents, giving rise to six distinct clades [7,13,14]. These clades are categorized by their geographic locations: Clade I (South Asian), Clade II (East Asian), Clade III (South African), Clade IV (South American), Clade V (Iran), and Clade VI (Singapore). The clades are genetically distinct, with variations among strains within the same clade differing by thousands of single-nucleotide polymorphisms. Each clade has very few unique single-nucleotide polymorphisms (SNPs), indicating that clonal expansion has occurred in this region [15]. Small differences among these SNPs led to clinically significant phenotypic variations such as body site tropisms, virulence, host colonization, outbreak potential, and antifungal resistance [4,13,16,17,18,19,20,21].

*C. auris* was first reported in the United States in 2016, with the initial seven cases occurring in New York (n = 3), Illinois (n = 2), Maryland (n = 1), and New Jersey (n = 1) between 2013 and 2016 [22]. Initially, *C. auris* infections in the U.S. were primarily imported; however, local transmission within healthcare settings has caused a significant rise in cases since its initial report. In 2018, clinical cases of *C. auris* were designated as a nationally notifiable condition, and as of 2023, screening cases also became notifiable [23]. The number of reported cases in the United States increased significantly, rising from 479 in 2019 to 1471 in 2021 [24]. Clinical cases rose annually, growing by 44% in 2019 and 95% in 2021. In 2021, colonization screening volume increased by over 80%, while screening cases surged by more than 200% [25]. The most rapid growth occurred between 2020 and 2021, when cases nearly tripled, reaching a total of 4401 [24,25]. In 2023, the U.S. reported 4514 new clinical cases of *C. auris*, continuing the upward trend since the emergence of the fungus [9]. Major outbreaks in America include New York State and Illinois, which both have over a thousand clinical cases as of April 2024 [9,19,21,26]. The data show a rising trend in *C. auris* colonization and clinical cases, primarily in long-term-care facilities, including long-term acute care and skilled nursing facilities with ventilators [26,27]. *C. auris* has been found in over half of the U.S. and is classified by the CDC as an urgent antimicrobial resistance threat [28].

As of December 2023, *C. auris* has been identified in 61 countries across six continents [29]. It has been reported across all subregions of Africa, with more than 2500 cases documented in the literature to date. Only six countries account for these reported cases [30]. Major outbreaks have been reported in the United Kingdom, Spain, and Italy, along with sporadic cases from several other European countries [31]. In certain healthcare settings in South Africa and India, *C. auris* has been responsible for as much as 25% and 40% of candidemia cases [32]. Outbreaks of *C. auris* bloodstream infections have been reported significantly in healthcare settings worldwide [1].

Nosocomial infections caused by *C. auris* have increased dramatically since their first report in South Korea in 2011, due to the organism’s ability to persist in environments, easily transmit between hosts, and proliferate rapidly with a clonal lifecycle [5]. From 2013 to 2018, 18 studies involving more than 200 patients reported nosocomial infections in more than 10 countries [33]. The COVID-19 pandemic significantly increased nosocomial infections caused by *C. auris* [25,34]. The mortality rate associated with *C. auris* infection rose from 50 to 65% in the pre-COVID-19 era to 60–80% during the COVID pandemic [5,35,36,37].

### Risk Factors

Contact with patients who carry *C. auris* or are in its environment poses a risk for colonization by *C. auris* [38]. The duration of contact necessary for the acquisition of *C. auris* from a colonized patient or environment is suggested to be as brief as four hours [39]. Additionally, invasive infections have been documented within 48 h of patient admission to intensive care units [40]. Multiple factors heighten the risk of developing *C. auris* infections, similar to those associated with other *Candida* species [1,41,42]. These factors include advanced age; the presence of indwelling medical devices, such as central venous or urinary catheters; specific comorbidities, including diabetes mellitus, neoplastic disease, and chronic kidney disease; the administration of total parenteral nutrition; reliance on mechanical ventilation; and the need for hemodialysis [2,20,22,27,39,41]. Additionally, individuals who are immunocompromised—due to conditions such as neutropenia, glucocorticoid therapy, or previous organ transplantation—are at heightened risk [20,35,39]. Other contributing elements are recent surgical procedures, severe cases of COVID-19, and recent exposure to broad-spectrum antibiotics or antifungal agents [39,40,41,42,43,44]. Collectively, these conditions create an environment conducive to the development of skin microbial communities that favor *C. auris* proliferation. Skin colonization is the most significant risk factor for developing candidemia [22].

## 4. Pathogenesis

### 4.1. Transmission

*C. auris* is highly contagious and spreads effectively from person to person [45]. The yeast has a high mortality rate attributed to the fact that infections are largely hospital acquired by other patients who carry C. auris as asymptomatic colonizers [46]. *C. auris* colonization can persist for months or even longer [47]. Identifying these asymptomatic colonized patients is vital for implementing contact precautions during surgeries or placing indwelling devices to prevent infection transmission.

### 4.2. Virulence Factors

Multiple factors contribute to the pathogenesis of *C. auris* infections, enhancing its survival in harsh environments and enabling disease in the host. Virulence factors may be strain-dependent. The genome of *C. auris* encodes various lytic enzymes, including secreted aspartyl proteases (SAPs), secreted lipases, and phospholipases [48]. Similarly to other *Candida* species, the SAPs in *C. auris* enhance their virulence by cleaving host proteins. These enzymes aid adhesion, biofilm formation, and tissue invasion while disrupting immune responses [48,49,50]. Hydrolases are the most commonly secreted enzymes in *C. auris*, representing 42% of its encoded enzymes [48]. Most *C. auris* strains have active hemolysin enzymes that efficiently sequester iron, promoting rapid growth and spread [51]. Additionally, lipases contribute to biofilm formation, host cell damage, and immune evasion [52]. *C. auris* phospholipases are generally less effective than those of *Candida albicans*, except for the CBS 12770 strain [53].

*C. auris* can evade the immune response using unique mechanisms. Examination of human neutrophil interactions with *C. auris* and *C. albicans* revealed that neutrophils primarily target and eliminate *C. albicans*, while *C. auris* significantly inhibits neutrophils by targeting neutrophil extracellular traps (NETs) to evade ingestion [54,55]. Further observations indicated that the outer mannan layer of the cell wall protects *C. auris* from phagocytic responses [56]. *C. auris* has been shown to lower macrophage glucose levels and lead to their death without triggering inflammasome responses [57]. High levels of β-1,2-linkages in the terminal mannan chains of *C. auris* were associated with differential IgG binding [58]. The IL-17 receptor signaling pathway restricts *C. auris* colonization on the skin of mice, as its disruption leads to increased fungal recovery from the skin [18].

Another key virulence factor is the strain-dependent ability of *C. auris* to form biofilms, which helps it adhere to surfaces and plastics. Transcriptome profiling of *C. auris* reveals that it upregulates adhesin proteins (CSA1, IFF4, PGA26, PGA52) essential for forming and maintaining biofilms [59]. The agglutinin-like sequence (ALS) proteins ALS1 and ALS5 in *C. auris* are thought to aid in biofilm adherence [59]. *C. auris* has fewer ALS and other adhesin genes than *C. albicans*, which may explain why *C. albicans* biofilms are more prevalent and robust [48,53]. The GPI-anchored cell wall genes related to adhesins were upregulated at every stage of biofilm formation. As the biofilm progressed to intermediate and mature stages, several efflux pump genes, including ATP-binding cassette (*ABC*) transporters (*CDR1, SNQ2, YHD3*) and major facilitator superfamily (*MFS*) transporters (*MDR1, RDC3*), were upregulated [59].

*C. auris* possesses two distinct forms: aggregating and non-aggregating. The aggregating form is particularly notable for its inability to release daughter cells after budding, leading to the formation of cell clusters that are resistant to detergent disruption [60]. This characteristic significantly enhances its survival in hospital environments, demonstrating its adaptability and resilience as a pathogen. Aggregative phenotypes noted in colonized patients form biofilms more effectively than non-aggregative phenotypes, making eradication efforts more difficult [61]. In vivo models show that non-aggregating isolates are more pathogenic than both aggregating isolates due to their ability to move around more effectively [48,60].

*C. auris* also shows phenotypic plasticity with three forms: typical yeast, filamentous-competent (FC) yeast, and fully filamentous [62]. This ability to switch morphologies in response to environmental conditions may enhance its virulence and infection potential. Filamentous forms of *C. auris* isolates show increased MICs in fluconazole and posaconazole from the typical yeast form [63]. The *C. auris* genome contains three genes homologous to the white–opaque regulator (WOR)1 of *Candida albicans*, which may regulate phenotypic switching in *C. auris* [64,65]. The CDC warns diagnostic facilities to be cautious with morphological features when screening for *C. auris* due to phenotypic switching. A better understanding of its phenotypes improves isolation success and reduces false-negative results.

*C. auris* has a unique stress resistance profile among pathogenic *Candida* species, largely due to its Hog1 stress-activated protein kinase (SAPK) [46]. This kinase enhances resistance to osmotic pressure, hydrogen peroxide (H_2_O_2_), sodium dodecyl sulfate (SDS), and other cell wall-damaging agents, contributing to the pathogen’s virulence [46]. *C. auris* metabolism favors respiration, as demonstrated by the heightened expression of glycolytic and sugar transporter genes during yeast growth [62]. This respiratory metabolism boosts adenosine triphosphate (ATP) production and reduces oxidative stress, enhancing its fitness in vivo and contributing to resistance against fluconazole [66]. Additionally, *C. auris* exhibits higher levels of ergosterol and structural lipids than *C. albicans*, which may affect its stress response and antifungal resistance [67]. Genes involved in iron transport and metabolism are significantly upregulated in the biofilm and filamentous forms of *C. auris*, highlighting the importance of iron acquisition [59,62] (Figure 1).

### 4.3. Resistance Factors

The high mortality rates linked to *C. auris* result mainly from its antifungal resistance [53]. Biofilm formation and the expansion of genes associated with drug resistance are the principal mechanisms driving antifungal resistance. *C. auris* biofilm formation leads to significant upregulation of glucan and mannan polysaccharides, resulting in a polysaccharide-rich matrix that sequesters approximately 70% of available triazole antifungals. This reduces drug penetration to the cells embedded within the biofilm and plays a key role in enhancing *C. auris* resistance to antifungal therapies [68,69]. *C. auris* isolates exhibiting biofilm formation demonstrated resistance to all tested antifungal agents, including fluconazole, echinocandins, and polyenes. In contrast, planktonic *C. auris* isolates were resistant solely to fluconazole [70]. Genetic studies of *C. auris* have revealed that resistance to azole and echinocandin antifungals is linked to mutations in the lanosterol 14-alpha-demethylase (ERG11) gene and the drug target 1,3-beta-glucan synthase (FSK1). All amphotericin B-resistant *C. auris* isolates have a novel amino acid substitution (G145D) in the ERG2 gene [71]. A separate study also found that a two-component signal transduction system, SSK1, and the mitogen-activated protein kinase HOG1 signaling pathway contribute to this resistance [72]. Efflux pumps such as the ATP-binding cassette (ABC) and major facilitator superfamily (MFS) also play a role in azole resistance. The gene CDR1 encodes the ABC efflux pump in *C. albicans*. In a study by Rybak, a gene analogous to CDR1 was identified in *C. auris*. When this gene was removed, this caused a decrease in itraconazole MIC by 2-fold [73].

## 5. Clinical Manifestations

Patients may be colonized with *C. auris* in the absence of any clinical signs or symptoms. The skin is the most common site of colonization, particularly in the nares, axilla, fingertips, and groin [21,39,41]. Skin colonization by *C. auris* is abnormal and is asymptomatic in over 90% of individuals [74]. The positive rates for screening sites in *C. auris*-positive patients are 80% for combined sampling of axilla and groin, 58% for bilateral nares alone, 50% for unilateral groin and axilla alone, 43% for unilateral nares, and 100% for combined bilateral axilla/groin and nares [75]. The current CDC recommendation is to use the axillae and groin regions as screening sites for *C. auris* [76]. A feature that distinguishes *C. auris* from other fungal pathogens is its high capacity to colonize skin, leading to widespread outbreaks in healthcare facilities via patient-to-patient transmission. *C. auris* forms dense biofilms under conditions that simulate sweat on the skin’s surface. These adherent biofilm communities explain the propensity of *C. auris* to colonize skin and persist in environmental conditions expected in the hospital setting [77]. Researchers discovered that patients with skin microbiomes dominated by Malassezia species had a lower risk of *C. auris* colonization. They identified more abundant bacteria, such as *Proteus mirabilis* and *Klebsiella pneumoniae*, in *C. auris*-colonized patients, while *Staphylococcus hominis* was more prevalent in those without *C. auris* [21]. *C. auris* differs from other *Candida* species in that it is not a commensal organism in the human gut, as it does not grow well in anaerobic or acidic environments [46].

The clinical presentation of a *C. auris* infection closely parallels that of other species within the *Candida* genus. Bloodstream infections have been the most observed and reported invasive infection caused by *C. auris* [8,22,39,40,41,78,79]. About 5% to 10% of patients who are colonized develop bloodstream infections [80]. The risk of developing *C. auris* candidemia exceeds 25% within 60 days following the initial detection of *C. auris* colonization [81]. It can spread hematogenously to distant anatomical sites and has been reported to cause myocarditis, pericarditis, meningitis, hepatosplenic infection, osteomyelitis, urinary tract infections, endophthalmitis, ear infections, and wound infections, as well as donor-derived disease following lung transplantation [78,80,82]. Mucosal infections, including oral thrush, esophageal candidiasis, and vulvovaginal candidiasis, are infrequent with *C. auris* [82,83]. Isolations from non-sterile body sites such as the lungs, urinary tract, skin and soft tissue, and genital apparatus may more likely represent colonization rather than infections [84]. The estimated in-hospital mortality rate for *C. auris* candidemia ranges from 30% to 70% [8,78,79,80,85]. Cases with high mortality rates involved patients with comorbidities and long-term healthcare exposure.

## 6. Diagnosis

### 6.1. Culture Presentation and Biochemical Techniques

*Candida auris* on Sabouraud dextrose agar (SAB) produces smooth white- or cream-colored colonies [1]. *C. auris* has several features which are distinct from its closest relatives. Slow and weak growth at 42 °C can occur with *C. auris* and *C. ruelliae*. No growth occurs at this temperature with *C.haemulonii*, *C. pseudohaemulonii,* and *C. heveicola* [1]. A further difference between *C. auris* and *C. ruelliae* is that *C. auris* does not assimilate galactose, l-sorbose, cellobiose, l-arabinose, ethanol, glycerol, salicin, or citrate as carbon sources. However, *C. ruelliae* can assimilate these carbon sources [1,57]. Additional testing is performed by detecting pseudohyphae. Out of all the *Candida* species similar to *C. auris,* only *C. auris* and *C. heveicola* do not produce pseudohyphae on cornmeal agar at 25 °C [1], though in rare and certain circumstances, *C. auris* can induce pseudohyphae-like forms. Growth with high amounts of salt in the medium can induce stress such as with YTD (yeast extract, tryptone, and dextrose) plus 10% NaCl. In addition to the depletion of heat shock proteins, this can produce pseudohyphae-like forms [86].

Traditional methods of identifying *C. auris* are not typically used in clinical laboratories due to the wide availability of Chromogenic media. Chromogenic media have added benefits of quickly distinguishing a mixed *Candida* species sample through different enzymatic reactions [87]. The appearance of *C. auris* on CHROMagar^TM^
*Candida* (Kanto Chemical Co., Tokyo, Japan) is typically white to pink, though some colonies can appear to be red or purple. This can help to rule out *Candida albicans, Candida tropicalis, Candida krusi, Candida kefyr*, and *Candida glabrata* as they have enzymatic reactions specific to their species, but this medium does not detect *C. auris* specifically. A new medium has come out called CHROMagar^TM^ Candida Plus, which gives C. auris a specific enzymatic reaction, resulting in a blue halo around the colonies. Even with this specific reaction, the identity of *C. auris* on this medium must be confirmed on another identification method [88]. Biochemical methods and MALDI-TOF (Bruker, MA, USA) are the typical confirmatory identification methods. Biochemical methods have identification challenges due to the close phylogenetic relationships with other *Candida* species (Figure 2). Some methods also misidentify with *Rhodotorula glutinis, Saccharomyces kluyveri*, and *Saccharomyces cerevisiae* [88,89]. Due to these potential misidentifications, as with Chromogenic media, it is also recommended to perform confirmatory testing with other methods [89]. The CDC has a helpful testing algorithm based on methods available to help prevent misidentification in laboratories [89,90].

Broth microdilutions from CLSI and EUCAST have been the standard for antifungal susceptibility testing for many years. Most of the systems mentioned use a variation of this during testing. However, there have been recent advances in using MALDI-TOF for susceptibility testing known as the MALDI Biotyper antibiotic susceptibility test rapid assay (MBT-ASTRA) method (MBT-ASTRA prototype software). This has the potential to reduce the time required to receive a susceptibility profile by a day. However, major research efforts need to be made to optimize this assay in order for it to receive approval from the FDA.

### 6.2. Molecular Techniques

Molecular techniques for *C. auris* identification have significant advantages over traditional phenotypic methods, which are prone to misidentification. Molecular methods, particularly those based on sequencing the D1–D2 region of the 28S ribosomal DNA or the internal transcribed spacer (ITS) region of the ribosomal cistron, are considered the most accurate methods for *C. auris* identification [1]. These sequencing techniques allow for precise species identification at the genetic level, overcoming the limitations of phenotypic methods. However, sequencing typically requires the isolation of a pure colony from traditional culture, a process that can be time-consuming and prone to failure, especially when fungal loads are low or mixed infections are present. Also, the high cost and complexity associated with sequencing isolates prevents most clinical laboratories from performing sequencing.

Many clinical laboratories have instead turned to PCR-based tests for the identification of *C. auris*. PCR-based tests allow for a more efficient and reliable approach by enabling rapid identification directly from clinical samples like blood or swabs. These tests can produce results within hours, often before culture results are available, and retain the high sensitivity and specificity associated with sequencing-based tests. The ability of PCR-based tests to rapidly detect small amounts of *C. auris* DNA without needing colony isolation has made it a preferred method in many healthcare settings, particularly in cases where timely diagnosis is critical for patient outcomes.

While PCR-based tests vary in complexity, two main types are commonly used in clinical laboratories: laboratory-developed tests (LDTs) and commercially available tests. LDTs are custom-designed in-house tests that can be tailored to the specific needs of a laboratory but still require specialized equipment and expertise to perform. In contrast, commercially available FDA-cleared/-approved tests such as the ePlex Blood Culture Identification Fungal Pathogen (BCID-FP) Panel (GenMark, Carlsbad, CA, USA) and the BioFire Blood Culture Identification 2 (BCID2) Panel (BioMerieux, France) offer a more rapid, standardized, and user-friendly approach. These panels are designed for ease of use with minimal technical expertise, providing automated results for a wide range of fungal pathogens from positive blood cultures, including *C. auris*. A more recent addition to commercially available, FDA-approved tests is the Simplexa *C. auris* Direct (DiaSorin, Italy), which is specifically designed to detect *C. auris* from axilla/groin surveillance samples in patients suspected of colonization. This test provides a highly sensitive, user-friendly, PCR-based solution that enhances the prevention and control of *C. auris* infections in healthcare facilities.

## 7. Treatment

Antifungal resistance is a growing threat that presents a major clinical challenge in treating *C. auris* infections due to its multidrug-resistant profile. About 90% of *C. auris* isolates in the United States have been resistant to fluconazole, about 30% have been resistant to amphotericin B, and less than 2% have been resistant to echinocandins [25]. Regional differences in *C. auris* susceptibility show that the Midwest has the lowest azole resistance, while the Mid-Atlantic region can have up to 85% resistance to amphotericin B [25]. Before 2020, echinocandin resistance was below 5% in all regions in the US. However, in 2021, the number of patients with echinocandin-resistant and pan-resistant isolates reportedly increased. Two echinocandin-resistant and pan-resistant *C. auris* outbreaks occurred among patients in shared healthcare environments with no prior echinocandin exposure, marking the first documented transmission of this resistance in US healthcare settings [91]. Among invasive candidiasis isolates in the 2022 SENTRY Antifungal Surveillance Program for *C. auris*, 82.1% of the isolates were resistant to fluconazole; 17.9% were resistant to amphotericin B; and 1.3% were resistant to caspofungin, anidulafungin, or micafungin. Rezafungin resistance was seen in 3.8% of the isolates, 17.9% of the isolates were resistant to fluconazole and amphotericin B, and no pan-drug resistance was seen [92].

The increasing resistance to triazole and amphotericin B has necessitated the recommendation of echinocandins as an empirical treatment before susceptibility testing results are available (Table 1). Adults and children over two months may be treated with echinocandins, specifically caspofungin or micafungin, while anidulafungin is limited to adults. Rezafungin has a long half-life and can be administered weekly. However, its main drawback is that, as an echinocandin, it is ineffective against isolates with FKS1 mutations. For patients who do not respond clinically, liposomal amphotericin B may be considered as an alternative. Amphotericin B deoxycholate is the recommended treatment for neonates and infants under two months of age [93].

Treatment for *C. auris* should be initiated only in the presence of clinical disease and should be avoided in patients colonized with *C. auris,* particularly when the organism is isolated from non-invasive sites such as the respiratory tract, urine, or skin. Most cases are managed on an individual basis, with guidance from susceptibility testing. It is highly recommended to consult an infectious disease specialist for optimal management and treatment strategies. Antifungal susceptibility testing should be performed in all clinical isolates, and patients should be closely monitored for acquired resistance while on treatment. Tentative breakpoints for antifungal resistance have been provided by CDC using the Clinical Laboratory Standard Institute (CLSI) method: MIC ≥ 32 μg/mL (fluconazole), MICs ≥ 2 μg/mL (amphotericine B), ≥4 μg/mL (anidulafungin), ≥2 μg/mL (caspofungin), and ≥4 μg/mL (micafungin), respectively. Fluconazole susceptibility can be a surrogate for the susceptibility assessment of other triazoles; occasionally, isolates resistant to fluconazole may respond to other triazoles including voriconazole; the decision to treat with another triazole will need to be made on a case-by-case basis [93].

Source control including removal of catheters with adequate drainage and/or debridement is an important part of therapy of candidiasis. The site of infection is key in selecting antifungal agents for invasive fungal infections. Echinocandins are not suitable for infections of the urinary tract and central nervous system (CNS) due to their poor therapeutic concentrations [94,95]. For urinary tract infections, amphotericin B, potentially with 5-flucytosine, is recommended [96]. Furthermore, empirical treatment for CNS infections with amphotericin B and 5-flucytosine has shown effectiveness, especially when optimized through sensitivity testing [96]. Duration of antifungal therapy should be guided by clinical response and the adequacy of source control. The Infectious Disease Society of America advises continuing antifungal treatment for two weeks after negative blood cultures in patients without metastatic complications (e.g., endocarditis, osteomyelitis, or suppurative thrombophlebitis) [96]. It is recommended to consult an ophthalmologist for a dilated eye examination in instances of candidemia.

In pan-resistant strains, where there is resistance to all three major classes of antifungals (echinocandins, amphotericin B, and azoles), evidence regarding the most appropriate therapy is lacking. In vitro studies on combinations of echinocandins with azoles or amphotericin B, as well as 5-flucytosine with other antifungal classes, demonstrated either synergy or indifference, with no antagonism observed. Variability among *C. auris* isolates was noted. The most effective combinations were azoles with echinocandins. Studies found no antagonism, with observed synergy or indifference among *C. auris* isolates [97]. In a separate study, flucytosine combined with amphotericin B, azoles, or echinocandins was found to be highly effective. Time–kill analysis showed each combination achieved over a 2 log10 reduction in growth, indicating fungicidal action [10].

New antifungal agents are being evaluated for treating invasive candidiasis caused by *C. auris*. *Ibrexafungerp* is a new antifungal agent effective against *C. auris*, even with echinocandin resistance. It is active against isolates with FKS1 mutations. An open-label study is ongoing to evaluate the effectiveness of ibrexafungerp for infections caused by *C. auris* [98].

*Fosmanogepix,* which is a prodrug of its active component, manogepix, belongs to a new class of antifungal medications. It works by inhibiting Gwt1, an enzyme that plays a crucial role in synthesizing glycosylphosphatidylinositol. This molecule is essential for anchoring mannoproteins to the cell wall and membrane, thereby altering their integrity and slowing fungal growth [98].

Vaccines have the potential to be a vital strategy in reducing the global impact of drug-resistant C auris. C auris has protein homologs of C albicans Als3p, a protein with adhesin and invasin properties important for host pathogenesis [99,100]. NDV-3A is a vaccine derived from the N-terminus of the Als3 protein, with aluminum hydroxide as an adjuvant, and has shown efficacy and safety against Candida species in pre-clinical and phase 1 clinical trials [101,102]. Vaccination of mice with NDV-3A generated antibodies against Als3p, which recognized C auris, inhibited its biofilm formation, and enhanced the ability of macrophages to kill the fungus [103]. This vaccination also produced strong T-cell immunity, protecting immunosuppressed mice from lethal infections of C auris. Additionally, NDV-3A improved the effectiveness of micafungin against C auris candidemia, resulting in better survival rates [104]. The next step is to conduct human trials to confirm the vaccine’s safety and efficacy for high-risk patients, particularly those with weakened immune systems.

## 8. Infection Control

*C. auris* can spread through contact with contaminated surfaces and fomites. *C. auris* frequently colonizes the skin, respiratory tract, and urinary tract and is shed from the skin into the environment, contaminating surfaces and equipment and causing transmission of infection through direct or indirect contact in hospital settings [11]. Contact from colonized individuals can also cause infection of *C. auris* via skin-to-skin contact [11]. Currently, there is no evidence of zoonotic transmission of *C. auris*, though it has been found to colonize dogs’ oral cavity [105]. *C. auris* colonization can persist indefinitely in patients. In a previous study, positive *C. auris* culture was identified during clinical or screening activities in healthcare settings, and when discharged to a community, the median time from initial *C. auris* identification to serially negative at assessments was 8.6 months [106]. Rapid detection of *C. auris* and prompt infection control measures are important to identify infected or colonized patients to prevent transmission.

Infection control recommendations for *C. auris* have been adapted from those for other pathogens, such as multidrug-resistant organisms (MDROs), which can quickly spread in healthcare settings. Improved infection control measures have decreased the transmission of other MDROs in healthcare settings. Alcohol-based hand sanitizer (ABHS) is the preferred method for hand hygiene. When hands exhibit visible dirt, cleaning them using soap and water is essential. Patients should be placed in private rooms to prevent the transmission of *C. auris*. In acute care and long-term acute care hospitals, it is necessary to implement contact precautions. Conversely, skilled nursing facilities should adopt enhanced barrier precautions to ensure optimal safety and infection control [11]. Patients with *C. auris* may be cohorted in the same room if private rooms are unavailable. However, patients with *C. auris* and different MDROs should not be cohorted together.

The CDC recommends maintaining contact precautions or enhanced barrier precautions throughout all inpatient healthcare stays, depending on the healthcare setting [11]. Patient rooms and areas in contact with them must be terminally cleaned and disinfected using appropriate disinfectants to ensure a safe environment. It is recommended to use List P from the United States Environmental Protection Agency (EPA), as it includes products with EPA-registered claims effective against *C. auris*. If List P is unavailable, you can use List K, which contains disinfectants effective against *Clostridioides difficile* spores [11,107,108]. Shared medical equipment should be cleaned and disinfected after each use.

Active surveillance and contact tracing must be conducted to identify other patients who may have been exposed to *C. auris* and to screen them for asymptomatic colonization. Screening should be based on local *C. auris* epidemiology and burden, epidemiologic linkages to cases, patient risk factors, and the purpose of screening [11]. 

Re-screening patients who are known to be infected or colonized with *C. auris* is not recommended. Patients who are colonized may remain colonized for extended periods, but this can occur intermittently. Screening of healthcare personnel is not recommended unless there is known or suspected transmission or other strong epidemiological links [11]. 

## 9. Conclusions

*C. auris* continues to spread rapidly and explosively in both colonization and infections throughout the globe. As with many infectious diseases, patients who are debilitated or immunocompromised are at the highest risk for severe illness due to *C. auris*. Echinocandins are recommended as initial treatment for adults before susceptibility testing results. They should only be used if there is clinical disease present, not for patients colonized with *C. auris.* This organism poses a challenge to therapy due to its ability to develop resistance through multiple mechanisms and mutations. Efforts undertaken by healthcare professionals in the domain of antimicrobial stewardship are essential for mitigating the development of increasingly resistant organisms. Barriers to antimicrobial stewardship include poor communication, misidentification, and lack of education. The limited effectiveness of many antifungal agents raises concerns about the urgent need for new treatments, as current options are scarce for combating this highly resistant fungus. Identifying colonized individuals and taking infection control precautions is the essential first step to help prevent the spread to critically ill patients. Emphasizing hand washing, isolation, and environmental measures is essential. Colonization is a prerequisite for developing life-threatening *C auris* infections, with bacteremia—which has a high mortality rate—being the most frequent. Colonized patients may remain so for long periods; thus, re-screening is not recommended. Screening of healthcare personnel is also not recommended unless there is known or suspected transmission or other strong epidemiological links. The development of current diagnostic capabilities has been slow, but improvements are being made. Molecular techniques are currently the preferred method for rapid and accurate identification, enabling appropriate treatment for this serious infection. *C auris* has the potential to become our next major threat to hospitalized individuals and needs to be viewed as a global threat to our patients. 

## Figures and Tables

**Figure 1 microorganisms-13-00652-f001:**
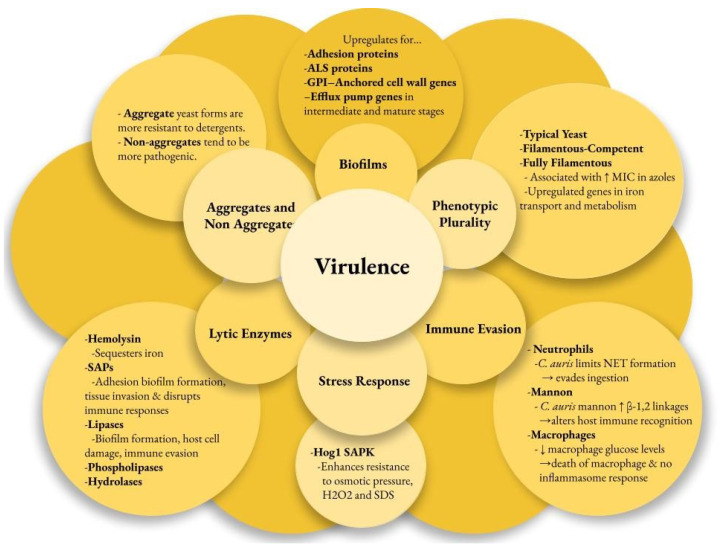
*Candida auris* virulence factors. ALS—agglutin-like sequence, GPI—glycosylphosphatidylinositol, MIC—minimum inhibitory concentration, NET—neutrophil extracellular traps, Hog—high osmolarity glycerol, SAPK—stress-activated protein kinase, SAP—secreted aspartyl proteases.

**Figure 2 microorganisms-13-00652-f002:**
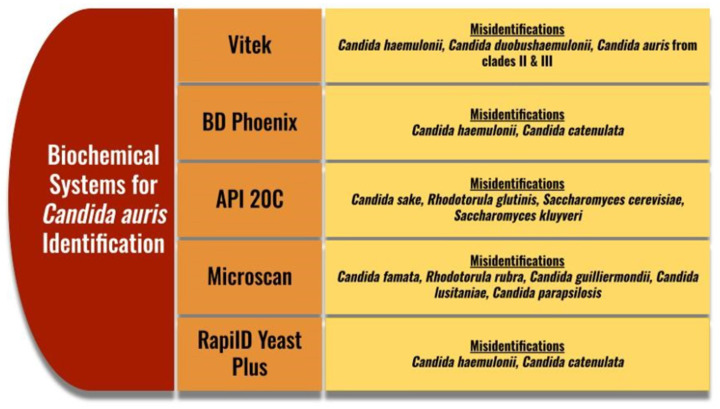
Common misidentifications for *Candida auris* on biochemical systems [88,89]. Vitek (BioMerieux, Craponne, France), BD Phoenix (BD Biosciences, NJ, USA), API 20C (BioMerieux, France), Microscan (Beckman Coulter, CA, USA, RapiID Yeast Plus (Remel, KS, USA).

**Table 1 microorganisms-13-00652-t001:** Initial treatment recommendation for *Candida auris*.

Antifungal	Adult Dose	Pediatric Dosing for ≥2 Months of Age	Neonates and Infants <2 Months of Age
Anidulafungin	Loading dose 200 mg IV, then 100 mg IV daily	Not approved	Not approved
Caspofungin	Loading dose 70 mg IV, then 50 mg IV daily	Loading dose 70 mg/m^2^/day IV, then 50 mg/m^2^/day IV (based on body surface area)	* Caspofungin—25 mg/m^2^/day IV (based on body surface area)
Micafungin	100 mg IV daily	2 mg/kg/day IV with option to increase to 4 mg/kg/day IV in children at least 40 kg	* Micafungin—10 mg/kg/day IV
Rezafungin	Loading dose 400 mg IV, then 200 mg IV weekly	Not approved	Not approved
Polyenes	Treatment not recommended initially	Treatment not recommended initially	Amphotericin B deoxycholate, 1 mg/kg daily (if unresponsive, consider liposomal amphotericin B, 5 mg/kg daily)

* In exceptional circumstances, where central nervous system involvement has been definitively ruled out, use of above echinocandins may be considered with caution: adapted from References [93,94].

## Data Availability

No new data were created or analyzed in this review article. Data sharing is not applicable to this article.

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
