# Peer review of "Candida auris: A Continuing Threat"

_microorganisms, 2025, doi:10.3390/microorganisms13030652_

Round 1

Reviewer 1 Report

Comments and Suggestions for Authors

Dear authors,

Thank you for the interesting research. I have some comments to improve your manuscript:

1.         Can three authors have the status of correspondence author? If so, e-mail addresses must be provided for all three.

2.         Line 61 - 'C. auris isolates...' - the name of the microorganisms should be in italics. Please check and correct this throughout the manuscript.

3.         In the introduction section, the purpose of the paper is missing. It must be stated.

4.        Figure 1 - please place the figure in the centre of the page, unfortunately the diagram has moved off the page.

5.         Figure 2 - please place the figure in the centre of the page, unfortunately the diagram has moved off the page.

6.         Table 1 - Polyenes - there is a blank row and a blank cell below. Please fill in the cells.

7.         I think a small section on the development of vaccines against C. auris could be added to increase the novelty.

8.         The manuscript lacks sections such as Author Contributions, Funding, Data Availability Statement, Conflicts of Interest. Add these sections according to the journal template.

Author Response

Reviewer 1 

Dear authors, 

Thank you for the interesting research. I have some comments to improve your manuscript: 

  1. Can three authors have the status of correspondence author? If so, e-mail addresses must be provided for all three.  

Didn't mean to add Dr.Saravoltaz as a third corresponding author only the two. 

  1. Line 61 - 'C. auris isolates...' - the name of the microorganisms should be in italics. Please check and correct this throughout the manuscript.

The document was reviewed and corrected with italics for all organisms, punctuation was also corrected in the references (lowercased the species name). 

  1. In the introduction section, the purpose of the paper is missing. It must be stated.

The paper's purpose was added in lines 42-44. 

  1. Figure 1 - please place the figure in the centre of the page, unfortunately the diagram has moved off the page.

Moved and resized. 

  1. Figure 2 - please place the figure in the centre of the page, unfortunately the diagram has moved off the page.

Moved and resized. 

  1. Table 1 - Polyenes - there is a blank row and a blank cell below. Please fill in the cells.

Deleted the row containing polyenes and put them all under as antifungal drugs instead of drug groups. 

  1. I think a small section on the development of vaccines against C. auris could be added to increase the novelty.

Added a few sentences on the subject. 

  1. The manuscript lacks sections such as Author Contributions, Funding, Data Availability Statement, Conflicts of Interest. Add these sections according to the journal template.

Added these sections according to the template. 

Reviewer 2 Report

Comments and Suggestions for Authors

Dear Authors,

please find the attached file.

Author Response

Reviewer 2 

The manuscript Candida auris: A Continuing Global Threat by Ashish Bhargava 1,2,*, Katherine Klamer 1*, Mamta Sharma 1, Daniel Ortiz 3, and Louis Saravoltaz 1,2,* has been revised. It focuses on the current situation of the Candida auris pathogen worldwide to provide guidance for further researches. After the revision of the manuscript, I do not recommend publishing this study in presented form for several reasons provided below: 

  1. The Introduction section should be supplemented with more detailed data on the distribution of C auris in different countries of the world, as the title of the article requires. However, the Authors should take into account recent publication (review) of Microorganisms https://doi.org/10.3390/microorganisms12050927, on the analysis of the same fungus strain.

Changed title. 

  1. It is not clear how the Authors selected the reference database and what the search strategy was? It should be discussed. 

We used a variety of applications such as google search, google scholar and searching within the journals themselves. We have discussed and will make changes moving forward. 

  1. 3. Please remove 4 Henry Ford Health - St. John Hospital, Detroit, MI from the affiliations or specify which author corresponds to this affiliation. 

Removed. 

  1. All Latin names of fungi and bacteria should be written in Italic. 

The document was reviewed and corrected with italics for all organisms, punctuation was also corrected in the references (lowercased the species name). 

  1. 5. P5: Figure 1 is too large. Please, reupload a more appropriate one. 

Moved and resized. 

  1. 6. [80] is not cited. 

Reference 80 was cited in line 277 of the original document “[8, 78, 79, 80, 81]’ 

  1. 7. P8: As for Figure 1, Figure 2 should be reduced. 

Moved and resized. 

  1. 8. P10: Table 1 needs to be presented more neatly. Next, The data of Rezafungin is not provided in [91]. Please check it and correct if necessary. The same for Polyenes. 

Deleted the row containing polyenes and put them all under as antifungal drugs instead of drug groups. Reference 91 had the incorrect reference, corrected reference was added, added reference for rezifungin. 

  1. 9. L407-409: mL instead of ml. 

Punctuation corrected. 

  1. 10. L408: anidulafungin instead of anidulofungin. 

Spelling corrected. 

11.L424: …[94]. 

Punctuation corrected. 

  1. 12. Paragraphs L436-445 and L441-445] should be cited. 

Citation was missing added reference 94. 

  1. 13. References [99] and [100] must be cited consecutively but not after [86] and [88].

Ordering of references was fixed.  

  1. 14. Ref[1]: Journal title should be included. 

Added 

  1. 15. Ref[4]: Chow, Nancy… 

Fixed 

  1. 16. Ref[5]: publication year should be included. 

Added 

  1. 17. Please check all references and unify the submission of journal titles: submit either abbreviated or full titles for all references.

Unified the reference formats. Added full titles for all references. 

Round 2

Reviewer 2 Report

Comments and Suggestions for Authors

Dear Authors,

the manuscript Candida auris: A Continuing  Threat  by Ashish Bhargava 1,2,* , Katherine Klamer 
1 1 , Mamta Sharma , Daniel Ortiz 3 , and Louis Saravoltaz 1,2, * was improved, but some inaccuracies were left. Please, correct them:

1- 4th affiliation was not removed. Please, remove it.

2- There is no clear explanation of how the references for the review were selected. It should be included in the manuscript.

3 - L39, 53, 242, 495: Refs should be cited at the end of the sentence before a period.

4 - L188: For H2O2, 2 in subscript.

5- L277: Ref [80] should be cited after Ref [79], but not after Ref [85]. please, renumber thereferences so that the referencese can be cited consecutively.

6 - L285: C. heveicola [1]. 

7 - L326: In the title of Figure 2: What is the meqaning of "a" of 87a, 88a? Explain or remove, please.

8 - In Table 1: 100 mg IV daily   - left alignment.

9 - Paragraph L446-458: References with A - is A required next to Ref numbers? If not, please remove the existing A.

After these revisions, the manuscript would be suitable for publication in Microorganisms.

Author Response

Dear Authors, 

the manuscript Candida auris: A Continuing  Threat  by Ashish Bhargava 1,2,* , Katherine Klamer  
1 1 , Mamta Sharma , Daniel Ortiz 3 , and Louis Saravoltaz 1,2, * was improved, but some inaccuracies were left. Please, correct them: 

1- 4th affiliation was not removed. Please, remove it. 

Now its removed 

2- There is no clear explanation of how the references for the review were selected. It should be included in the manuscript. 

We selected a variety of sources searching using PubMed, Google Search, Google Scholar, and individual journal sites (such as ASM) from a range of dates from 2009 to present day. 

3 - L39, 53, 242, 495: Refs should be cited at the end of the sentence before a period. 

I fixed those and multiple others I found within the document. 

4 - L188: For H2O2, 2 in subscript. 

Subscripted the 2 in H2O2. 

5- L277: Ref [80] should be cited after Ref [79], but not after Ref [85]. please, renumber thereferences so that the referencese can be cited consecutively. 

6 - L285: C. heveicola [1].  

Fixed orientation of the period. 

7 - L326: In the title of Figure 2: What is the meqaning of "a" of 87a, 88a? Explain or remove, please. 

Used when moving the references. Removed. 

8 - In Table 1: 100 mg IV daily   - left alignment. 

Removed indentation, aligned to the left. 

9 - Paragraph L446-458: References with A - is A required next to Ref numbers? If not, please remove the existing A. 

Used when moving the references. Removed. 

After these revisions, the manuscript would be suitable for publication in Microorganisms.